# Tire Ground Rubber Biodegradation by a Consortium Isolated from an Aged Tire

**DOI:** 10.3390/microorganisms10071414

**Published:** 2022-07-14

**Authors:** Sarelia M. Castañeda Alejo, Kevin Tejada Meza, María R. Valderrama Valencia, Armando J. Arenazas Rodríguez, Christian J. Málaga Espinoza

**Affiliations:** 1Departamento Académico de Arquitectura e Ingenierías Civil y del Ambiente, Universidad Católica de Santa María—UCSM, Urb. San José, San Jose S/n, Yanahuara, Arequipa 04000, Peru; sarelia.castaneda@ucsm.edu.pe (S.M.C.A.); aarenazas@ucsm.edu.pe (A.J.A.R.); cmalaga@ucsm.edu.pe (C.J.M.E.); 2Departamento Académico de Farmacia, Bio-Química y Biotecnología, Universidad Católica de Santa María—UCSM, Urb. San José, San Jose S/n, Yanahuara, Arequipa 04000, Peru; marov52@ucsm.edu.pe

**Keywords:** *Delftia tsuruhatensis*, next generation sequencing, shredded tire rubber, Sturm test, Schiff test, biodegradation

## Abstract

Rubber is a natural product, the main car tire component. Due to the characteristics acquired by this material after its vulcanization process, its degradation under natural conditions requires very long times, causing several environmental problems. In the present work, the existence of a bacterial consortium isolated from a discarded tire found within the Socabaya River with the ability to degrade shredded tire rubber without any chemical pretreatment is explored. Taking into consideration the complex chemical composition of a rubber tire and the described benefits of the use of pretreatments, the study is developed as a preliminary analysis. The augmentative growth technique was used, and the level of degradation was quantified as a percentage through the analysis of microbial respiration. Schiff’s test and the use of comparative photographs of scanning electron microscopy (SEM) were also used. The consortium using next generation genetic sequencing was analyzed. A 4.94% degradation point was obtained after 20 days of experimentation, and it was found that the consortium was mostly made up with *Delftia tsuruhatensis* with 69.12% of the total genetic readings of the consortium and the existence of 15% of unidentified microbial strains at the genre level. The role played by the organisms in the degradation process is unknown. However, the positive results in the tests carried out show that the consortium had action on the shredded tire, showing a mineralization process.

## 1. Introduction

Rubber is a natural product from the *Hevea brasiliensis* [1] tree. Its characteristics make this material have a large number of usefulness, being the most important raw material in tire manufacture. Year after year, the global demand for tires grows. By 2021, the Peruvian automotive fleet incorporated 175,000 new cars [2]. This translates into a demand for approximately 700,000 new tires, being 40% higher than the demand in 2020. This increase, added to the impossibility of reusing the material in the manufacturing process [3] and their long periods of degradation in natural conditions, causes an accumulation process and the occurrence of other problems, such as the proliferation of pests [4] or tire combustion [5].

Because of these characteristics, car tires are considered to be recalcitrant pollutants, a category that they share along with plastic and its derivatives [6]. There are recovery strategies based on the use of discarded tires as raw material, mainly for the development of handicrafts or in the manufacture of synthetic soccer grass [7,8,9]. However, the number of tires that can be reduced by this process is very limited, and at the same time they can cause other problems due to the chemical components they contain, observing the presence of polycyclic aromatic hydrocarbons (PAH) and heavy metals in recycled products from tires [10].

In this sense, the research focuses on the development of processes that allow elimination of the component or reduction to simpler materials with some usefulness if possible; processes such as pyrolysis [11], mechanical recovery [12] and cryogenic recovery [13] emerge as ideal candidates. However, the high technological costs and the level of technical training of the operators as a condition are limitations in its application on a global scale.

Technologies based on the use of biological processes are a complementary tool to the processes described above. There are also economical strategies which could work as an analogue to some chemical processes. Earlier studies have shown the ability of some microorganisms for rubber degradation, mostly belonging to organisms isolated from contaminated soil or water. Actinomycetal or Xanthomonadal order bacteria are the most common. These are represented by organisms such as Nocardia [14,15], several *Gordonia* strains [16,17,18] and *Xanthomonas* sp. [19] that have been extensively described in the literature. Additionally, others such as Streptomycetaceae [20,21] and Pseudomonadales [22] have also been shown to have the ability to degrade rubber. Likewise, rubber biodegradation studies with the use of microbial consortia have been previously carried out [23,24]. 

Little is known about the metabolic pathways involved in the rubber biodegradation process and today the bacterial degradation process is still slow and complicated to achieve. Therefore, finding a bacterium or consortium with better performance is very important. The present study analyzes the possible ability of rubber biodegradation (without any pretreatment) of an isolated microbial consortium in a seasonal river from the city of Arequipa.

## 2. Materials and Methods

### 2.1. Sample Collection

A fragment of a submerged tire was extracted from the swampy area of the Socabaya River, Arequipa, Peru. The selected sample met the following criteria: presence of surface fractures, surface wear and colonization of plant and animal organisms within the concavities of the tire. The fragment was transported to the laboratory keeping the cold chain at 4 °C ± 0.5. 

### 2.2. Microorganisms’ Acclimatization with Rubber Biodegradation Potential

The tire fragment was suspended in a mineral salt medium (MSM) described by Kanwal et al. [25] (KH_2_PO_4_ 0.04 g/L, K_2_HPO_4_ 0.5 g/L, NaCl 0.1 g/L, CaCl_2_.2.H_2_O 0.002 g/L, (NH_4_)_2_SO_4_ 0.02 g/L, MgSO_4_.7.H_2_O 0.2 g/L, FeSO_4_ 0.001 g/L, at pH 7) for a period of 7 moths. During the growth period, 100 μL of medium from the degradation flask were sampled to mix with 100 μL chloroform and Schiff’s reagent in order to demonstrate the presence of metabolites related to rubber degradation (aldehydes or ketones) [21,26].

### 2.3. Rubber Degradation System (Sturm Test)

The construction of the system was based on the model proposed by Shah et al. [26], and in accordance with the parameters proposed in ISO 9439 [27]. The experiment was carried out in triplicate with a control for each test. Each degradation flask had an initial bacterial concentration of 1.34 × 10^7^ Bacteria/mL. Ground tire rubber (donated by the local retreading company RELINO S.A) was sifted to reach 850 µm particle size and then added to the degradation system until it reached a 0.5% weight volume concentration. The set-up growth conditions were 150 RPM in orbital agitation, 35 °C and constant aeration.

### 2.4. CO_2_ Produced Calculation

The calculation of the CO_2_ concentration was carried out by applying the equations indicated by ISO 9439 [27]. At the beginning of the experiment the initial CO_2_ concentration was calculated, and the evolution was measured periodically (every 2 days at the beginning and then every 5 days). The collected samples were titrated with 0.1 M HCl in order to quantify the amount of CO_2_ trapped. The CO_2_ production and the biodegradation percentage also known as CO_2_ evolution were analyzed and calculated.

### 2.5. SEM Image Analysis

The SEM analysis was used to confirm the surface weakening after the biological treatment. The treated samples with the microbial consortium were subjected to SEM analysis after washing with 2% (*v*/*v*) aqueous SDS and distilled water for a few minutes and rinsed with 70% ethanol to remove cells after 20 incubation days [28]. The sample was covered with gold and analyzed under a high-resolution scanning electron microscope (LS-15; Carl Zeiss, Jena, Germany).

### 2.6. Bacterial Consortium Molecular Identification Capable of Degrading Rubber

To identify the organisms present in the consortium, a metagenomic analysis was done using the Next Generation Sequencing (NGS) technique with next generation sequencing (NGS) equipment of the 16S rRNA gene. As a first step, the library was prepared by fragmentation of the DNA chain, and then amplified by PCR. Generation clusters were then performed to amplify the base signal to meet the signal requirements for the sequence. Finally, sequencing was performed to compare the chains obtained with the genetic libraries [29].

## 3. Results and Discussion

### 3.1. Bacterial Growth Curve

The consortium growth curve is showed in Figure 1. The exponential phase takes place from day 1 to day 4. The stationary phase began on day 5 and lasted 20 days. There was a decrease point after day 30 and a small recovery on day 35. 

### 3.2. Rubber Degradation

A CO_2_ maximum production was observed around day 20. This corresponds to the growth curve stationary phase of the consortium (Figure 2). Neither medium or carbon source (ground tire rubber) was added. 

### 3.3. SEM Analysis

From the scanning electronic microscopy (SEM), a change in the rugosity of the rubber surface can be observed, showing an increase in roughness in the degraded matrix. (Figure 3B).

### 3.4. Consortium Characeristics

The metagenomic study of the consortium found in the trial (Figure 4) shows that the microorganism with the greatest presence was *Delftia tsuruhatensis* (69.12%), which becomes a new species reported for this type of rubber degradation analysis. This was followed by a group of unidentified organisms constituting 15.69%, which becomes a candidate to carry out studies for identification; it may be a new species not reported so far since its sequences have not been found in the database of the main germplasm banks. This was followed by *Delftia laustris* (3.78%) and *Acidovorax*
*wohlfahrti* (1.10%); however, it should be highlighted that 15.69% of the consortium readings corresponded to unidentified organisms at species level. The bacterial orders found are shown in Table 1.

The results show the presence of a *microviridae* viral strain which is the phiX174 micro virus enterobacteria bacterial phage. This result is due to the technique used for the Illumina sequencer DNA detection. This consists of adding from 2% to 5% of the sequence belonging to this phage in a controlled manner in order to establish a more homogeneous bacterial DNA distribution [30]. Therefore, these results obtained will not be taken into consideration for this study, but they will be presented for not altering the final percentages. 

## 4. Discussion

The present study aimed to demonstrate the existence of a native microorganism consortium from the City of Arequipa, capable of degrading ground tire rubber without any chemical pre-treatment. 

As presented in Table 2, a microorganism consortium was able to successfully reduce 4.94% of the rubber matrix through the proposed method [27].

Studies performed by Berekaa et al. [17] where the *Gordonia* sp. cepa *Kb2* bacterial strain was used showed a 4% degradation efficiency after 40 days of operation on latex gloves without any pretreatment and Linos et al. [31] applied *Pseudomonas aeruginosa AL98* for the degradation of latex gloves without any pretreatment which showed a 26% efficiency after 6 weeks. Considering that the tire chemical composition is more complex and presents a larger number of components than latex gloves, the result obtained in the current experience indicates that it is likely that the bacterial consortium has a better performance when compared to previous experiences, being able to be optimized to reach higher degradation values. 

The bacterial species with a higher presence in the consortium correspond to the *Delftia tsuruhatensis* and *Delftia lacustris* species. No previous reports of their participation in rubber degradation have been found. Both bacteria are commonly found in rivers and ecosystems with contaminated water. Environmental uses of these microorganisms have been described. *Delftia tsuruhatensis* was reported in terephthalates degradation [32]. At the same time, both microorganisms have had an important role in the metabolism and reduction of aromatic pollutants, such as anilines [33] or bisphenol S [34]. Rubber tires contain polycyclic aromatic hydrocarbons (PAH) which have been found in recycled rubber tires [10]. These components could be the reason why both organisms are the ones presented the most in the consortium. A study was performed by Marchut-Mikołajczyk et al. [35] where an organism known for its ability to degrade anilines successfully degraded rubber tire. This supports the importance of PAH degrader organisms in rubber tire degradation. This could be a pretreatment to be applied along with biological desulfurization. The slow biodegradation observed in the test may be due, as indicated by Arutchlevi et al. [36], to the fact that the final degradation of the polymer can take hundreds of years, due to the fact that elements such as additives, antioxidants and stabilizers are added to commercial polymers that may be toxic to microorganisms or may slow down the rate of biodegradation. Either the presence of Actinomicetales and Xanthomonadales order previously reported in rubber tire degradation, or the positive reaction of Schiff’s reactive indicate the presence of compounds derived from tire degradation, such as aldehydes and ketones. This means that the microorganism action in the consortium was complementary, encompassing a larger number of chemical components inside the rubber of tires. 

It is not possible to experimentally ensure the role that the additional unreported organisms had in rubber degradation nor to corroborate the performance of *Delftia tsuruhatensis*. The study was not focused on analyzing derivatives of PAH degradation and therefore no methodologies were applied dedicated to the measurement of additional components to those described in the literature as derivatives of rubber degradation. In the same way, the results obtained from the genetic analysis of the consortium could be affected by the working time of the consortium, which could be located in an aged stage derived from the process of degradation and acclimatization of the organisms. The role of *Delftia tsuruhatensis* in the process is not clear which means that more detailed action is expected to be studied. Also, the produced metabolites analysis is expected to enlarge after degradation and determine the chemical components of the tire being attacked by the action of this microorganism. 

## 5. Conclusions

Microorganisms capable of degrading rubber in terrestrial-aquatic ecosystems were found in Socabaya River in the City of Arequipa. It was also determined that the *Delftia tsuruhatensis* strain, besides being the most abundant microorganism in the isolated consortium (69.12%), plays a significant role in the degradation process of rubber without pretreatment. Additionally, new species were found that had not previously been reported relative to their role in the degradation of rubber, some of which could not be identified at the species level. The consortium obtained a 4.94% ± 0.624 rubber biodegradation percentage. This is a similar value to the ones reported by other authors. These values can be even more encouraging due to the time and complexity of the matrix used (rubber without pretreatment). Electronic microscopy was a useful tool to corroborate the degradation performed by the consortium. Complex organic degradation alternatives of matrixes, such as rubber, are limited and the use of microorganisms is an eco-friendly and efficient solution. 

## Figures and Tables

**Figure 1 microorganisms-10-01414-f001:**
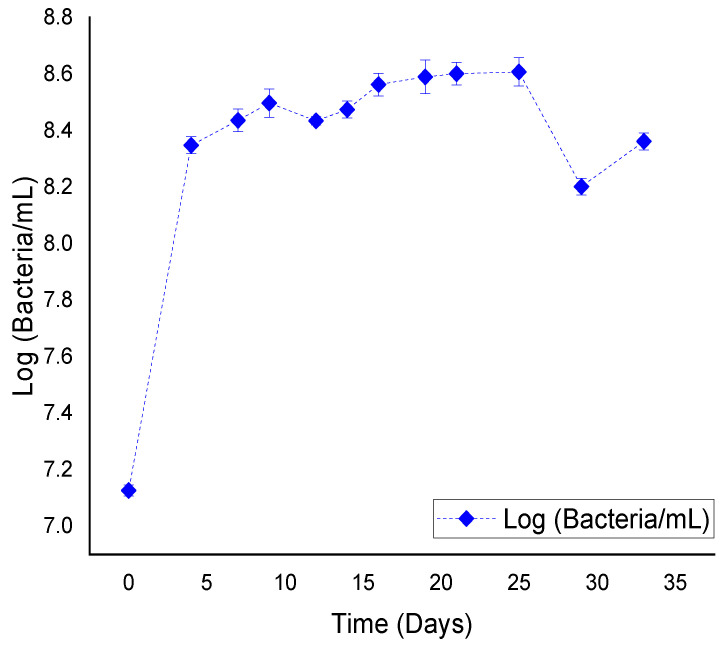
Growth curve of the microbial consortium isolated from submerged tire segments from a swampy area on the Socabaya River, Arequipa.

**Figure 2 microorganisms-10-01414-f002:**
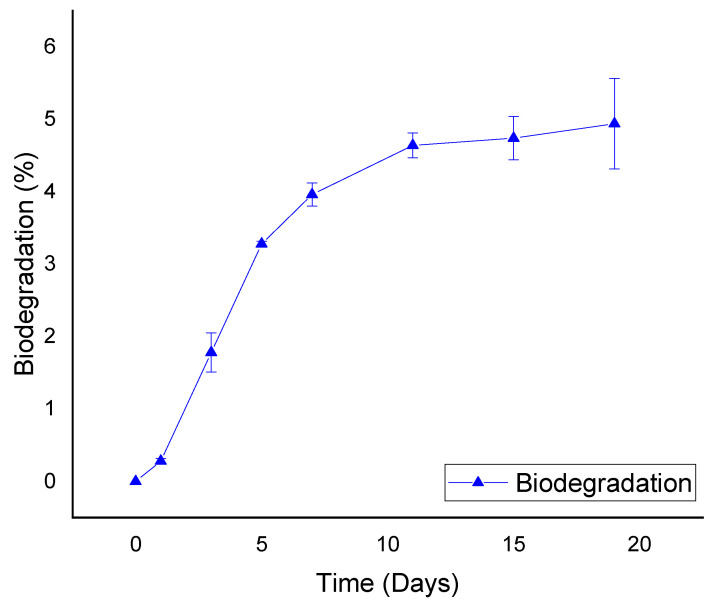
Percentage of rubber degradation.

**Figure 3 microorganisms-10-01414-f003:**
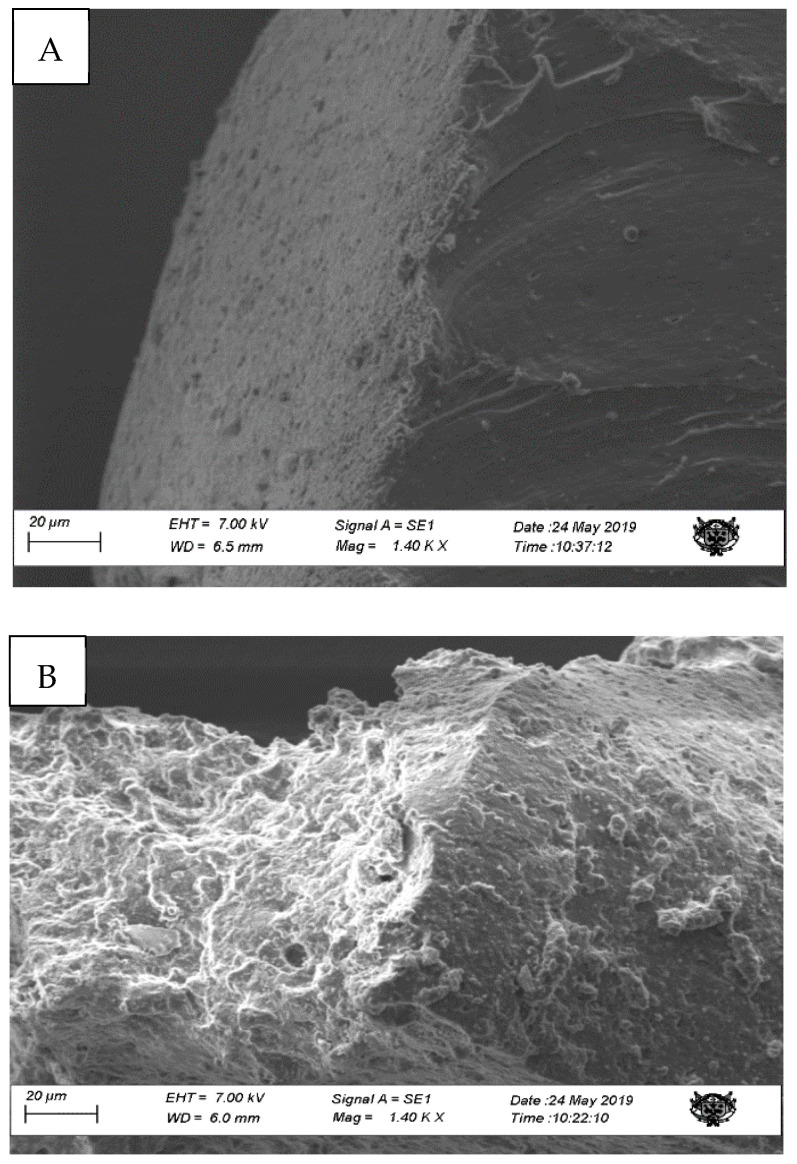
(**A**) Non-degraded rubber matrix, (**B**) degraded matrix—1.40 KX.

**Figure 4 microorganisms-10-01414-f004:**
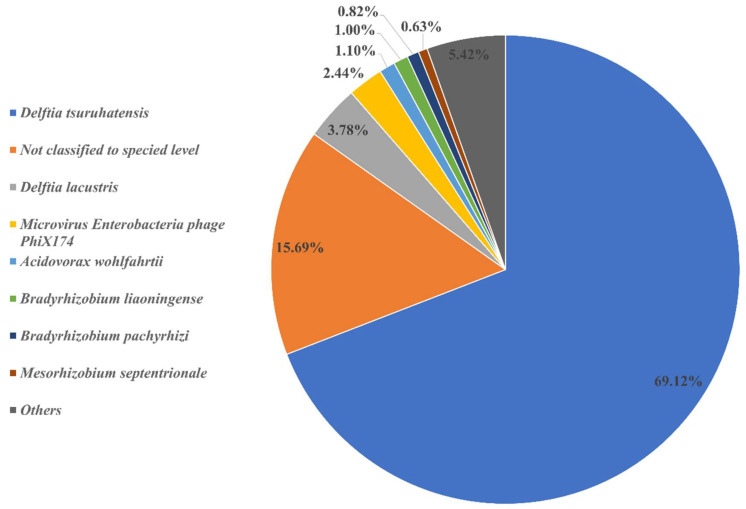
Native species found in the consortium.

**Table 1 microorganisms-10-01414-t001:** Identification by orders of rubber degradation in the consortium.

Clasification	%Total Readings *
Burkholderiales	82.65%
Rhizobiales	9.42%
Microviridae	2.44%
Sphingomonadales	1.90%
Non Clasified at level Order	0.79%
Actinomycetales	0.64%
Xanthomonadales	0.57%
Enterobacterial	0.24%

* 69 categories were identified; however, the table only shows the first 8 out of 69.

**Table 2 microorganisms-10-01414-t002:** Comparison of the degradation percentage.

Type of Culture	Type of Degraded Rubber	Biodegradation Percentage	Reference
**Consortium**	Vulcanized rubber (tire)	4.96% (20 days)	Our result
***Gordonia* sp. *strain Kb2* **	Latex glove without treatment	4% (40 days)	[17]
	Latex glove treated with chloroform	10% (40 days)	
	Latex glove treated with ketone	15% (40 days)	
** *Pseudomonas aeruginosa* **	Concentrated natural rubber	36% (40 days)	[31]
	Synthetic rubber	21% (40 days)	
	Latex glove	26% (40 days)	

## Data Availability

Not applicable.

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
