# Peer review of "Tire Ground Rubber Biodegradation by a Consortium Isolated from an Aged Tire"

_microorganisms, 2022, doi:10.3390/microorganisms10071414_

Round 1
Reviewer 1 Report
The S.M.C. Alejo et al. manuscript is devoted to the study of the processes of destruction of tire ground rubber under the influence of microorganisms and the study of the microbial community. In general, the article corresponds to the profile of the journal Microorganisms, but needs to be revision.
In the abstract, it is indicated that the degradation took place over 20 years. Is it so? Maybe 20 days? Tsuruhatensis should be written with a small letter.
The authors are asked to describe the sample preparation in more detail. How many samples were used, in what volume, what size were the particles.
The authors are asked to describe in more detail the method of microbial profiling.
Line 118. In the text, after the link to Figure 2, there is a link to Figure 4 immediately. And the link to Figure 3 is later. Please check the order.
In general, this article can be accepted as a preliminary study, but I would like to see a more complete description of the materials and methods
Reviewer 2 Report
Dears, I have only a few suggestions:
Improve the quality of figures 2, 3, and, 5;
Make a detailed check of the English;
Make the numerical decimals consistent in the text.
